# Towards Principled Representation Learning for Entity Alignment

## Abstract

Knowledge graph (KG) representation learning for entity alignment has recently received great attention. Compared with conventional methods, these embedding-based ones are considered to be robuster for highly-heterogeneous and cross-lingual entity alignment scenarios as they do not rely on the quality of machine translation or feature extraction. Despite the significant improvement that has been made, there is little understanding of how the embedding-based entity alignment methods actually work. Most existing methods rest on the foundation that a small number of pre-aligned entities can serve as anchors to connect the embedding spaces of two KGs. But no one investigates the rationality of such foundation. In this paper, we define a typical paradigm abstracted from the existing methods, and analyze how the representation discrepancy between two potentially-aligned entities is implicitly bounded by a predefined margin in the scoring function for embedding learning. However, such a margin cannot guarantee to be tight enough for alignment learning. We mitigate this problem by proposing a new approach that explicitly learns KG-invariant and principled entity representations, meanwhile preserves the original infrastructure of existing methods. In this sense, the model not only pursues the closeness of aligned entities on geometric distance, but also aligns the neural ontologies of two KGs to eliminate the discrepancy in feature distribution and underlying ontology knowledge. Our experiments demonstrate consistent and significant improvement in performance against the existing embedding-based entity alignment methods, including several state-of-the-art ones.

## 1 Introduction

Knowledge Graphs (KGs), such as DBpedia (Auer et al., 2007) and Wikidata (Vrandečić & Krötzsch, 2014), have become crucial data resources for many AI applications. Although a large-scale KG offers structured knowledge derived from millions of facts in the real world, it is still incomplete by nature, and the downstream applications are always demanding for more knowledge. To resolve this issue, the task of entity alignment (EA) is proposed, which exploits the potentially-aligned entities among different KGs to facilitate knowledge fusion and exchange.

Recently, embedding-based entity alignment (EEA) methods (Chen et al., 2017; Zhu et al., 2017; Wang et al., 2018; Guo et al., 2019; Ye et al., 2019; Wu et al., 2019; Sun et al., 2020a; Fey et al., 2020) have been prevailing in this area. Their common idea is to encode semantics into embeddings and estimate the similarities by embedding distance. During this process, a small number of aligned entity pairs (a.k.a., seed alignment) are required as supervision data to align (or merge) the embedding spaces of KGs. These methods either learn an alignment function $f_a$ to minimize the difference between two entity embeddings in each seed (Wang et al., 2018), or directly map aligned entities to one embedding vector (Sun et al., 2017). Meanwhile, they also leverage a shared scoring function $f_s$ to encode semantics into representations, such that two underlying aligned entities that connect to respective sides of a seed shall have similar characteristics in their feature expression.

Although the effectiveness of current EEA methods are empirically demonstrated (Sun et al., 2020b), little efforts have been made on the theoretical analysis. In this paper, we fill this gap by formally defining a paradigm leveraged by the current methods. We show that the representation discrepancy of an underlying aligned entity pair is bounded in an indirect way by a margin $\lambda$ in the scoring

function $f_s$. Unfortunately, we further find that this margin-based bound cannot be set as tight as expected, causing that little constrain can be put on the entities with few neighbors.

To mitigate the above problem, we propose neural ontology driven entity alignment (abbr., NeoEA) , in which the entity representations are optimized jointly with a neural ontology. An ontology (Baader et al., 2005) is usually comprised of axioms that define the legitimate relationships among entities and relations. Those axioms make a KG *principled* (i.e., constrained by rules). For example, an "Object Property Domain" axiom in OWL2 (Baader et al., 2005) claims the valid head entities for a specific relation (e.g., the head entities of relation "birthPlace" should be in class "Person"), and it thus determines the head entity distributions of this relation. The neural ontology in this paper, however, is reversely deduced from the entity distributions. We expect to align the high-level neural ontology to diminish the discrepancy of feature distributions, as well as ontology knowledge, between two KGs.

The main contributions of this paper are threefold:

- We define the paradigm of the current EEA methods, and demonstrate that the embedding discrepancy in each potential alignment pair is implicitly bounded by the margin in the scoring function. We show that this bound cannot be as tight as we expect.

- We propose NeoEA to learn *KG-invariant* as well as *principled* representations by aligning the neural axioms of two KGs. We prove that minimizing the difference can substantially align their corresponding ontology-level knowledge, without the assumption about the existence of real ontology data.

- We conducted experiments to verify the effectiveness of NeoEA with several state-of-the-art methods as baselines. The results show that NeoEA can consistently and significantly improve the performance of the EEA methods.

## 2 EMBEDDING-BASED ENTITY ALIGNMENT

### 2.1 METHODOLOGY

We first summarize the common paradigm employed by most existing EEA methods (Chen et al., 2017; Sun et al., 2017; Zhu et al., 2017; Sun et al., 2018; Wang et al., 2018; Pei et al., 2019a; Guo et al., 2019; Wu et al., 2019; Ye et al., 2019; Sun et al., 2020a):

**Definition 1** (Embedding-based Entity Alignment). *The input of EEA is two KGs $\mathcal{G}_1 = (\mathcal{E}_1, \mathcal{R}_1, \mathcal{T}_1)$, $\mathcal{G}_2 = (\mathcal{E}_2, \mathcal{R}_2, \mathcal{T}_2)$, and a small subset of aligned entity pairs $\mathcal{S} \subset \mathcal{E}_1 \times \mathcal{E}_2$ as seeds to connect $\mathcal{G}_1$ with $\mathcal{G}_2$. An EEA model consists of two neural functions: an alignment function $f_a$, which is used to regularize the embeddings of pairwise entities in $\mathcal{S}$; and a scoring function $f_s$, which scores the representations based on the joint triple set $\mathcal{T}_1 \cup \mathcal{T}_2$. EEA estimates the alignment score of an arbitrary entity pair $(e_i^1, e_j^2)$ by their geometric distance $d(\mathbf{e}_i^1, \mathbf{e}_j^2)$, where $\mathbf{e}_i^1$, $\mathbf{e}_j^2$ denote the embeddings of $e_i^1$, $e_i^2$ respectively.*

It is worth noting that, the existing EEA methods have different settings in relation seed alignment. Some works (Chen et al., 2017; Zhu et al., 2017) assume that all aligned relation pairs are known in advance. Others (Sun et al., 2017; 2018) suppose that the number of relations is much smaller than that of entities, i.e., $|\mathcal{R}| \ll |\mathcal{E}|$, which means that the training data for aligning relations is sufficient. In this paper, we do not explore the details of relation seed setting. We assume that the relation representations for a well-trained EEA model are aligned.

The existing works have explored the diversity of $f_a$. The pioneering work MTransE (Chen et al., 2017) proposed to learn a mapping matrix to cast an entity representation $\mathbf{e}_i^1$ to the feature space of $\mathcal{G}_2$. SEA (Pei et al., 2019a) and OTEA (Pei et al., 2019b) extended this approach by leveraging adversarial training to learn the projection matrix. Recently, a simpler yet more efficient choice was widely-used, which directly maps $(e_i^1, e_i^2) \in \mathcal{S}$ to one embedding vector $\mathbf{e}_i$ (Sun et al., 2017; Zhu et al., 2017; Trsedya et al., 2019; Guo et al., 2019). Also, researchers (Wang et al., 2018; Pei et al., 2019a; Wu et al., 2019) started to leverage a softer way to incorporate seed information, in which the distance between entities in a positive pair (i.e., supervised data in $\mathcal{S}$) is minimized, while that referred to the negative one will be enlarged. As the most common choice, we consider $f_a$ as

Euclidean distance between two embeddings, such that the corresponding alignment loss can be written as follows:

$$\mathcal{L}_a = \sum_{(e_i^1, e_i^2) \in \mathcal{S}} ||\mathbf{e}_i^1 - \mathbf{e}_i^2|| + \sum_{(e_{i'}^1, e_{j'}^2) \in \mathcal{S}^-} ReLU(\alpha - ||\mathbf{e}_{i'}^1 - \mathbf{e}_{j'}^2||), \tag{1}$$

where $\mathcal{S}^-$ denotes the sampled set of negative pairs. $\alpha$ is the minimal margin allowed between entities in each negative entity pair.

On the other hand, the scoring function $f_s$ can be also designed diversely. Most methods (Chen et al., 2017; Sun et al., 2017; Pei et al., 2019a) choose TransE as their scoring function, i.e., $f_s(e_i, r, e_j) = ||\mathbf{e}_i + \mathbf{r} - \mathbf{e}_j||, (e_i, r, e_j) \in \mathcal{T}_1 \cup \mathcal{T}_2$. The corresponding loss is:

$$\mathcal{L}_s = \sum_{\tau \in \mathcal{T}_1 \cup \mathcal{T}_2} ReLU(f_s(\tau) - \lambda) + \sum_{\tau' \in \mathcal{T}_1^- \cup \mathcal{T}_2^-} ReLU(\lambda - f_s(\tau')), \tag{2}$$

where $\mathcal{T}_1^-$ and $\mathcal{T}_2^-$ are negative triple sets. $\mathcal{L}_s$ is a margin-based loss in which the distance $d(\mathbf{e}_i + \mathbf{r}, \mathbf{e}_j)$ in a positive triple should at least be smaller than $\lambda \geq 0$, while larger than $\lambda$ for negative ones. Note that, the negative triples are usually generated by randomly replacing the head or tail entity of a positive triple. If we only look at the replaced entity, minimizing the above loss can be also understood as randomly pushing entities away from this entity. This phenomena has also been studied in Wang & Isola (2020).

Additionally, some graph neural network (GNN) based methods (Wang et al., 2018; Sun et al., 2020a; Wu et al., 2019) do not directly optimize $f_s$. They encode the semantic information inside graph convolution. Therefore, the output vectors of GNN will be regarded as entity embeddings to feed into $f_a$. For example, Ye et al. (2019); Wu et al. (2019); Sun et al. (2020a) leverage TransE as scoring function in the aggregation of relational neighbors.

## 2.2 UNDERSTANDING EEA

We illustarte how an EEA model works by an example. Let $(e_x^1, e_y^2) \in \mathcal{G}_1 \times \mathcal{G}_2$ be a potentially-aligned entity pair. Each entity in this pair has only one neighbor, connected by the same relation $r^1 = r^2$. We assume that their neighbors are actually a pair of entities $(e_i^1, e_i^2) \in \mathcal{S}$. Therefore, if an EEA model is well-trained and almost optimal, we should have $\mathbf{e}_i^1 = \mathbf{e}_i^2$ (as $\mathcal{L}_a$ is minimized) and $\mathbf{r}^1 = \mathbf{r}^2$ (denoted by $\mathbf{r}$ for simplicity). According to Equation 2, we have:

$$||f_s(\mathbf{e}_x^1, \mathbf{r}, \mathbf{e}_i^1)|| \approx ||f_s(\mathbf{e}_y^2, \mathbf{r}, \mathbf{e}_i^2)|| \leq \lambda. \tag{3}$$

Take the scoring function of TransE as $f_s$, we then derive:

$$||\mathbf{e}_x^1 + \mathbf{r} - \mathbf{e}_i^1|| \leq \lambda, \quad ||\mathbf{e}_y^2 + \mathbf{r} - \mathbf{e}_i^2|| \leq \lambda. \tag{4}$$

As $\mathbf{e}_i^1 = \mathbf{e}_i^2$, we can conclude that:

**Proposition 1** (Discrepancy Bound). *The representation difference of two potentially-aligned entities is bound by $\epsilon$, which is proportional to the hyper-parameter $\lambda$:*

$$||\mathbf{e}_x^1 - \mathbf{e}_y^2|| \leq \epsilon \propto \lambda. \tag{5}$$

The above proposition suggests that decreasing the value of margin $\lambda$ will tighten the feature discrepancy of entities in the underlying aligned entity pairs. However, we soon find that $\lambda$ cannot be set as small as we want.

We consider a more complicated yet realistic example, where each entity in $(e_x^1, e_y^2)$ has a considerable number of neighbors. We denote the corresponding triple sets of $e_x^1, e_y^2$ as $\mathcal{T}_{e_x}^1, \mathcal{T}_{e_y}^2$, respectively. In this setting, a well-trained EEA model should satisfy that:

$$\forall \tau \in \mathcal{T}_{e_x}^1 \cup \mathcal{T}_{e_y}^2, \ ||f_s(\tau)|| \leq \lambda. \tag{6}$$

Evidently, TransE with a small margin is not sufficient to fully express the semantics contained in $\mathcal{T}_{e_x}^1 \cup \mathcal{T}_{e_y}^2$, which has already been explored by previous works (Trouillon et al., 2016; Kazemi & Poole, 2018; Sun et al., 2019). Some empirical statistics (Sun et al., 2018) also illustrate such results. However, enlarging the margin $\lambda$ will bring significant variance between $\mathbf{e}_x^1$ and $\mathbf{e}_y^2$.

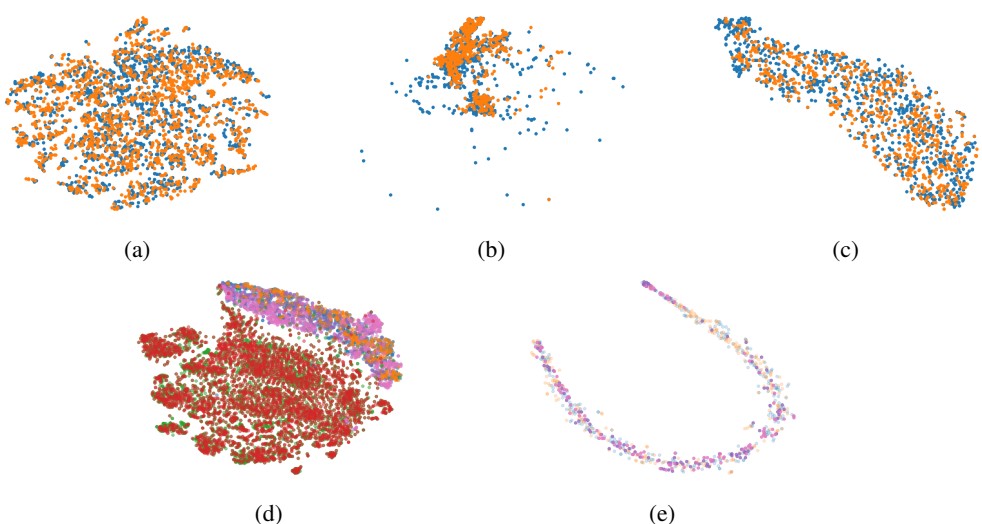

Figure 1: Example of different feature distributions. (a) Overall entity feature distributions of two KGs, i.e., $\mathbb{A}_E$. Blue points denote entities in $\mathcal{G}_1$ and orange ones are entities in $\mathcal{G}_2$. The two distributions are nearly uniformly distributed and almost aligned (based on the EEA model RDGCN (Wu et al., 2019)). (b) The head entity feature distributions of relation "genre". The two distributions are only aligned partially. (c) Head entity feature distributions conditioned on "genre", i.e., $\mathbb{A}_{E_h|r_i}$ (based on NeoEA with RDGCN as EEA model, the same below). Two conditioned distributions are aligned as expected. (d) The head entity distributions conditoned on three different relations: "genre" (colors: <blue,orange>), "writer" (colors: <purple,pink>), "brithPlace" (colors: <green,red>). The distributions corresponding to the first two relations are overlapped, while a clear decision boundary between them and the last one is observed. (e) Triple feature distributions conditioned on relations "artist" (colors: <blue,orange>) and "musicalArtist" (colors: <purple,pink>), respectively. The distributions referred to sub-relation "musicalArtist" are covered by those corresponded to "artist".

On the other hand, if the scoring function does not belong to the TransE family, e.g., it is neural-based like ConvE (Dettmers et al., 2018) or composition-based like ComplEx (Trouillon et al., 2016), both of which are fully expressive (Kazemi & Poole, 2018). In this case, entities with a large number of neighbors can be correctly modeled, while those with only few neighbors are less constrained. Therefore, those models allow even more diversity between $\mathbf{e}_x^1$ and $\mathbf{e}_y^2$. We believe this is why they performed badly in EA task (Guo et al., 2019; Sun et al., 2020b).

In short, most existing works adopt the above *implicit* strategy to learn cross-KG representations for EA, which makes them struggled in balancing between the bound and the expressiveness. In this paper, we explore a new direction to *explicitly* align the feature distributions of two KGs, which ensures the embeddings tight and expressive.

## 3 NEURAL ONTOLOGY

### 3.1 NEURAL AXIOM AND NEURAL ONTOLOGY ALIGNMENT

In real-world KGs, entities and their relation triples conform with the axioms in ontologies (Baader et al., 2005). Similarly, we call the feature distributions "neural axioms", as aligning them also allows us to regularize the entity embeddings at a high level.

We start by defining the basic neural axiom:

**Definition 2** (Basic Neural Axiom).

$$\mathbb{A}_E = \{\mathbf{e} \,|\, e \sim \mathcal{E}\}. \tag{7}$$

Aligning the basic neural axioms $\mathbb{A}_E^1$ and $\mathbb{A}_E^2$ of two KGs is trivial, as we can take the advantages of existing domain adaptation methods (Ben-David et al., 2010; Ganin & Lempitsky, 2015a;b; Courty

et al., 2017; Shen et al., 2018), which aims to learn domain-invariant representations for various tasks. We consider the adversarial learning based ones (Ganin & Lempitsky, 2015a; Shen et al., 2018). In this way, a KG discriminator is leveraged to distinguish entity representations of $\mathcal{G}_1$ from those of $\mathcal{G}_2$ (or vice versa), while the embeddings will try to confuse the discriminator. Therefore, the same semantics in two KGs shall be encoded in the same way into the embeddings to fool the discriminator.

Specifically, if we regard two KGs $\mathcal{G}_1$, $\mathcal{G}_2$ as two different domains, and their embedding vectors as "learnable features", we can align the above axioms by an empirical Wasserstein distance based loss (Arjovsky et al., 2017; Shen et al., 2018):

$$\mathcal{L}_{\mathbb{A}_E} = \mathbb{E}_{\mathbb{A}_{E^1}}[f_w(\mathbf{e})] - \mathbb{E}_{\mathbb{A}_{E^2}}[f_w(\mathbf{e})], \tag{8}$$

where $f_w$ is the learnable domain critic that maps the embedding vector to a scalar value. As suggested in (Arjovsky et al., 2017), the empirical Wasserstein distance can be approximated by maximizing $\mathcal{L}_{\mathbb{A}_E}$, if the parameterized family of $f_w$ are all 1-Lipschitz.

However, from Figure 1a, we observe that the trained embeddings are nearly uniformly distributed in the feature space, which we can also derive from Equation 1 and Equation 2. Recall that the alignment loss $\mathcal{L}_a$ consists of two terms. The first is

$$\sum_{(e_i^1, e_i^2) \in \mathcal{S}} ||\mathbf{e}_i^1 - \mathbf{e}_i^2||, \tag{9}$$

which aims to minimize the difference of embeddings for each positive pair. The cardinality of $\mathcal{S}$ is usually small. But this contrastive requires a large size of negative samples, which means that $||\mathcal{S}|| \ll ||\mathcal{S}^-||$. Therefore, the model more focuses on the second term

$$\sum_{(e_{i'}^1, e_{j'}^2) \in \mathcal{S}^-} ReLU(\alpha - ||\mathbf{e}_{i'}^1 - \mathbf{e}_{j'}^2||), \tag{10}$$

of which the main target is to randomly push the embeddings of different entities away from each other. Furthermore, $\mathcal{L}_s$ is also a contrastive loss, and has a similar effect on maximizing the pairwise distance between each positive entity and its corresponding sampled negative ones. Therefore, we conclude that:

**Proposition 2** (Uniformity). *The entity embeddings tend to be uniformly distributed in feature space as an EEA model is optimized.*

The above proposition suggests that only aligning the basic axioms may be insufficient to facilitate EEA. Hence, we propose conditional neural axioms which are more specific and expressive.

### 3.2 CONDITIONAL NEURAL AXIOM

Conditional neural axioms describe the entity (or triple) feature distributions under specific semantic conditions.

**Definition 3** (Conditional Neural Axioms).

$$\begin{aligned}
\mathbb{A}_{E_h | r_i} &= \{\mathbf{e} \,|\, \mathbf{r}_i, \; e \sim \{e \,|\, \forall e', (e, r_i, e') \in \mathcal{T}\}\} \\
\mathbb{A}_{E_{h,t} | r_i} &= \{(\mathbf{e}_h, \mathbf{e}_t) \,|\, \mathbf{r}_i, \; (e_h, e_t) \sim \{(e_h, e_t) \,|\, (e_h, r_i, e_t) \in \mathcal{T}\}\}
\end{aligned} \tag{11}$$

where $\mathbb{A}_{E_h | r_i}$ denotes the head entities feature distribution conditioned on the relation embedding $\mathbf{r}_i$, the similar to $\mathbb{A}_{E_{h,t} | r_i}$ (we reduce $(\mathbf{e}_h, \mathbf{r}_i, \mathbf{e}_t) | \mathbf{r}_i$ to $(\mathbf{e}_h, \mathbf{e}_t) | \mathbf{r}_i$ for simplicity).

Numerous methods are proposed to process the neural conditioning operation, ranging from addition and concatenation (Mirza & Osindero, 2014; Wang et al., 2014; Yang et al., 2015), to matrix multiplication (Lin et al., 2015a; Ji et al., 2015; Nguyen et al., 2016). Comparing with elaborating this operation, we value more on its common merit, which can be understood as projecting the entities to a relation-specific subspace (Wang et al., 2014; Lin et al., 2015a; Nguyen et al., 2016). Hence, the corresponding feature distributions conditioned on different relation embeddings become discriminative, rather than almost uniformly distributed in original feature space (Lin et al., 2015a).

Furthermore, conditional neural axioms capture high-level ontology knowledge.

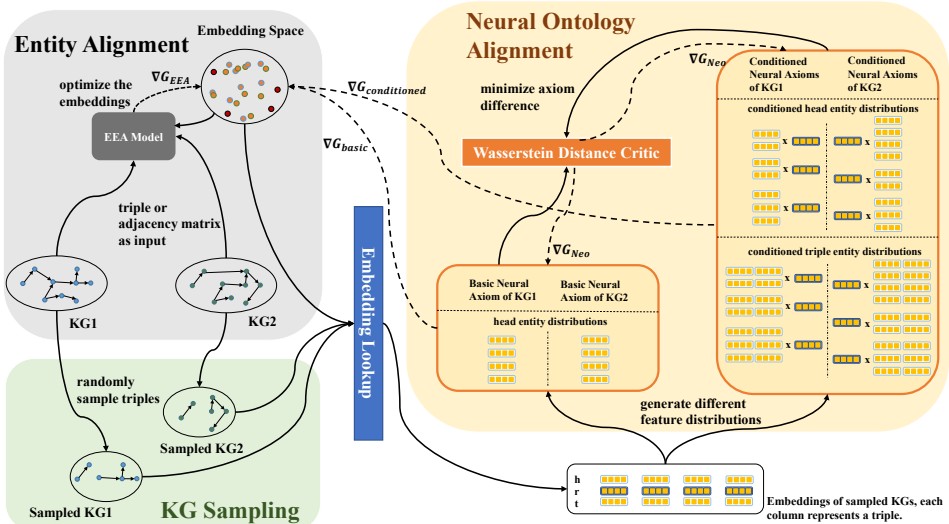

Figure 2: The architecture of NeoEA. The solid lines denote the forward propagation, while the dotted lines represents the backward propagation. The architecture consists of three decoupled modules: (1) Entity alignemnt module, in which the EEA model is unaware of the existence of other modules. (2) KG sampling module, in which we sample sub-KGs to replace the whole KGs for efficiency. (3) Neural ontology alignment module, in which the discrepancy between each pair of neural axioms is estimated by Wasserstein-distance critic and minimized by gradient descent.

**Theorem 1** (Expressiveness). *Aligning the conditional neural axioms minimizes the discrepancy of two KGs at ontology level.*

*Proof.* See Appendix A for details. We take $\mathbb{A}_{E_h|r_i}$ as an example, which can summarize the empirical "Object Property Domain" axiom of $r_i$ in OWL2 (Baader et al., 2005). Supposed there exists such an axiom that states the head entities of $r_i$ should belong to some specific class $c$ (e.g., only head entities under class "Person" have the relation "birthPlace"). We further suppose that there exists a classifier $f_c(\mathbf{e}) \in [0, 1]$, such that $f_c(\mathbf{e}_j) = 1$ if head entity $e_j$ belongs to class $c$, and 0 otherwise. Then, with the knowledge of the given axiom, one may derive the following rule:

$$\forall e \in \{e \,|\, \forall e', (e, r_i, e') \in \mathcal{T}_1 \cup \mathcal{T}_2\},\ f_c(\mathbf{e}) = 1, \tag{12}$$

which is equivalent to:

$$\mathbb{E}_{\mathbb{A}^1_{E_h|r_i}}[f_c(\mathbf{e})] = \mathbb{E}_{\mathbb{A}^2_{E_h|r_i}}[f_c(\mathbf{e})] = 1, \tag{13}$$

both of which means that all head entities of $r_i$ in either KG should be correctly classified to $c$. Then, we have:

$$\mathbb{E}_{\mathbb{A}^1_{E_h|r_i}}[f_c(\mathbf{e})] - \mathbb{E}_{\mathbb{A}^2_{E_h|r_i}}[f_c(\mathbf{e})] = 0. \tag{14}$$

In fact, we do not have such knowledge about $r_i$ and class $c$, instead we can leverage a neural function $f_{c'}(\mathbf{e}|\mathbf{r}_i)$ to empirically estimate $f_c$. In this way, $\mathbb{A}^s_{E_h|r_i}$ and $\mathbb{A}^t_{E_h|r_i}$ are supposed to be aligned to minimize the loss correponding to the above rule. Therefore, we deduce this problem back to a similar form to Equation 8, i.e.,

$$\mathcal{L}_{\mathbb{A}_{E_h|r_i}} = \mathbb{E}_{\mathbb{A}^1_{E_h|r_i}}[f_{c'}(\mathbf{e}\,|\,\mathbf{r}_i)] - \mathbb{E}_{\mathbb{A}^2_{E_h\,|\,r_i}}[f_{c'}(\mathbf{e}|\mathbf{r}_i)], \tag{15}$$

which suggests that aligning the above conditional neural axioms can minimize the discrepancy of potential "Object Property Domain" axioms between two KGs.

$\square$

**Example 1** (OWL2 axiom: ObjectPropertyDomain). *As shown in Figure 1b and Figure 1c, we assume that the head entity of relation "genre" are under class "Work of Art" (although it does not exist in the dataset). It is clear that the head entity feature distributions are only partially algined in Figure 1b, while those in Figure 1c are matched well.*

*In Figure 1d we illustrate a more complicated example. The head entities of relations "genre" and "writer" mainly belong to "Work of Art", which show overlapped distributions (blue-orange, pink-purple) in the figure. By contrast, there exists a clear decision boundary between them and the distributions conditioned on relation "birthPlace" (red-green), as the head entities of relation "birthPlace" are under class "Person".*

**Example 2** (OWL2 axiom: SubObjectPropertyOf). *We consider two relations "musicalArtist" and "artist" as example, where the former one is the sub-relation of the later one. In Figure 1e, the triple distributions conditioned on "musicalArtist" (pink-purple) are covered by those conditioned on "artist" (orange-blue).*

### 3.3 ARCHITECTURE OF NEOEA

We illustrate the overall structure of NeoEA in Figure 2. It can be divided into three modules:

**Entity Alignment.** This module aims at encoding the semantics of KGs into embeddings. Almost all existing EEA models can be used here, no matter what the input data looks like (triples or a adjacency matrix).

**KG Sampling.** For each KG, we choose to sample a sub-KG to estimate the data distributions of neural axioms. Comparing with separately sampling candidates for each axiom, it is more efficient especially when the size of KGs get bigger, because we only sample once at each iteration.

**Neural Ontology Alignment.** As aforementioned, for each pair of representation distributions, we align them by minimizing the empirical Wasserstein distance. Please see Appendix B for the detailed implementation of neural ontology alignment.

## 4 EXPERIMENTS

In this section, we empirically verify the effectiveness of NeoEA by a series of experiments, with several state-of-the-art methods as baselines.

### 4.1 IMPLEMENTATION

We illustrate the implementation of NeoEA in Algorithm 1. The whole framework is based on the OpenEA project (Sun et al., 2020b), which includes the implementations of latest EEA methods. Specifically, we implemented neural ontology as an external module, based on which we modified only the initialization of the original project. In this sense,, the EEA methods were unaware of the existence of neural ontologies. Furthermore, we kept the optimal hyper-parameter settings in OpenEA to ensure fair comparison.

We selected several best-performing and representative methods as our baselines:

- BootEA (Sun et al., 2018), a TransE-based EEA model with only structure data.
- SEA (Pei et al., 2019a), a TransE-based model with both structure and attribute data.
- RSN (Guo et al., 2019), an RNN-based EEA model with only structure data.
- RDGCN (Wu et al., 2019), a GCN-based model with both structure and attribute data.

The data distributions of some previous benchmarks such as JAPE (Sun et al., 2017) and BootEA (Sun et al., 2018) are clearly different from those of real-world KGs, which means that conducting experiments on those benchmarks cannot reflect the realistic performance of an EEA model (Guo et al., 2019; Sun et al., 2020b). Therefore, we consider the latest benchmark (Sun et al., 2020b), which consists of four sub-datasets, with two different density settings. Specifically, "D-W", "D-Y" denote "DBpedia (Auer et al., 2007)-WikiData (Vrandečić & Krötzsch, 2014)", "DBpedia-YAGO (Fabian et al., 2007)", respectively. "EN-DE" and "EN-FR" denote two cross-lingual datasets, both of which

are sampled from DBpedia. "V1" denotes the sampled KGs having the similar distributions as the original KGs, while "V2" denotes the sampled KGs with doubled density. For detail statistics, please refer to Sun et al. (2020b).

---

**Algorithm 1** NeoEA

---

1: **Input:** two KGs $\mathcal{G}_1$, $\mathcal{G}_2$, the alignment seed set $\mathcal{S}$, the EEA model $\mathcal{M}(f_s, f_a)$, number of steps for NeoEA $n$;
2: Initialize all variables;
3: **repeat**
4:    **for** $i := 1$ **to** $n$ **do**
5:       Sample sub-KGs from respective KGs $\mathcal{G}_1$, $\mathcal{G}_2$;
6:       Compute the Wasserstain distance based loss $\mathcal{L}_w$ for each pair of neural axioms;
7:       Optimize the Wasserstain distance critic $f_w$ by maximizing $\mathcal{L}_w$.
8:    **end for**
9:    Sample sub-KGs from respective KGs $\mathcal{G}_1$, $\mathcal{G}_2$;
10:    Compute Wasserstain distance based loss $\mathcal{L}_w$ for each pair of neural axioms;
11:    Compute the losses $\mathcal{L}_r$, $\mathcal{L}_s$ of the EEA model $\mathcal{M}$;
12:    Optimize the EEA model and embeddings by minimizing $\mathcal{L}_r$, $\mathcal{L}_s$, $\mathcal{L}_w$;
13: **until** the alignment loss on validation set converged.

---

## 4.2 EMPIRICAL COMPARISONS

The main results are shown in Table 1, from which we find that: (1) The performance of four baseline models varied from different datasets, but all of them gained improvement with NeoEA. (2) The performance improvement on SEA and RDGCN was more significant than that on BootEA and RSN, as both BootEA and RSN are not typical EEA models. BootEA has a sophisticated bootstrapping procedure, which may be difficult to be injected with NeoEA. RSN tries to capture long-term dependencies among entities and relations. The complicated objective may be conflict with NeoEA more or less. However, on some datasets (e.g., EN-DE, V1), we still observe relatively significant improvement. Therefore, we believe the performance of these two models can be further refined through a joint hyper-parameter turning with NeoEA, which we leave to future work.

Table 1: Entity alignment results (5-fold cross-validation).

| | Models | V1-Original | | | V1-NeoEA | | | V2-Original | | | V2-NeoEA | | |
|---|---|---|---|---|---|---|---|---|---|---|---|---|---|
| | | H@1 | H@5 | MRR | H@1 | H@5 | MRR | H@1 | H@5 | MRR | H@1 | H@5 | MRR |
| EN-FR | BootEA | .507 | .718 | .603 | .521 | .733 | .617 | .660 | .850 | .745 | .665 | .853 | .749 |
| | SEA | .280 | .530 | .397 | **.320** | **.584** | **.443** | .360 | .651 | .494 | .375 | **.666** | .508 |
| | RSN | .393 | .595 | .487 | .399 | .597 | .490 | .579 | .759 | .662 | .583 | .760 | .666 |
| | RDGCN | .755 | .854 | .800 | .775 | .868 | .817 | .847 | .919 | .880 | **.864** | .933 | **.896** |
| EN-DE | BootEA | .675 | .820 | .740 | .676 | .820 | .740 | .833 | .912 | .869 | .834 | .916 | .870 |
| | SEA | .530 | .718 | .617 | **.586** | **.766** | **.668** | .606 | .779 | .687 | **.637** | **.800** | **.712** |
| | RSN | .587 | .752 | .662 | .600 | .759 | .673 | .791 | .890 | .837 | .794 | .892 | .839 |
| | RDGCN | .830 | .895 | .859 | .846 | .908 | .874 | .833 | .891 | .860 | .849 | .902 | .874 |
| D-W | BootEA | .572 | .744 | .649 | .579 | .753 | .658 | .821 | .926 | .867 | .822 | .926 | .869 |
| | SEA | .360 | .572 | .458 | **.389** | **.608** | **.490** | .567 | .770 | .660 | **.588** | **.784** | **.677** |
| | RSN | .441 | .615 | .521 | .450 | .624 | .530 | .723 | .854 | .782 | .729 | .858 | .787 |
| | RDGCN | .515 | .669 | .584 | .527 | .671 | .592 | .623 | .757 | .684 | .632 | .760 | .690 |
| D-Y | BootEA | .739 | .849 | .788 | .756 | .859 | .797 | .958 | .984 | .969 | .958 | .984 | .969 |
| | SEA | .500 | .706 | .591 | **.549** | **.752** | **.638** | .899 | .950 | .923 | **.917** | **.959** | **.936** |
| | RSN | .514 | .655 | .580 | .522 | .663 | .588 | .933 | .974 | .951 | .935 | .976 | .953 |
| | RDGCN | .931 | .969 | .949 | .941 | .972 | .955 | .936 | .966 | .950 | .940 | .970 | .953 |

The results improved most are boldfaced.

## 4.3 ABLATION STUDY

We designed an ablation study which is expected to empirically prove some claims in Section 3. We choose the current state-of-the-art model RDGCN as our baseline. As shown in Table 2, "Full"

Table 2: Results of ablation study based on the best-performing model RDGCN, on V1 datasets.

| Models | EN-FR | | | EN-DE | | | D-W | | | D-Y | | |
|---|---|---|---|---|---|---|---|---|---|---|---|---|
| | H@1 | H@5 | MRR | H@1 | H@5 | MRR | H@1 | H@5 | MRR | H@1 | H@5 | MRR |
| Full | **.775** | **.868** | **.817** | **.846** | **.908** | **.874** | **.527** | **.671** | **.592** | **.941** | **.972** | **.955** |
| Partial | .771 | .863 | .813 | .840 | .900 | .871 | .523 | .669 | .590 | .936 | .971 | .952 |
| Basic | .755 | .853 | .799 | .827 | .895 | .858 | .512 | .656 | .578 | .931 | .969 | .948 |
| Original | .755 | .854 | .800 | .830 | .895 | .859 | .515 | .669 | .584 | .931 | .969 | .949 |

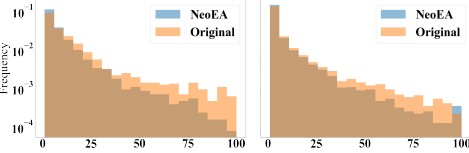

Figure 3: Normalized histgrams of alignment rankings on EN-FR, V1 (left, long-tail entities; right, popular entities).

Table 3: Average ranking improvement.

| Datasets | Overall | Popular | Long-tail |
|---|---|---|---|
| EN-FR | 63.5 | 36.9 | 116.7 |
| EN-DE | 13.0 | 8.1 | 23.4 |
| D-W | 43.5 | 34.5 | 61.3 |
| D-Y | 119.3 | 59.2 | 214.2 |

denote NeoEA with full set of neural axioms. "Partial" denotes NeoEA that removed the conditional triple axioms. We further removed the conditional entity axioms from "Partial" to construct "Basic", and the last one "Original" denotes the original EEA model. From the results we observe that: (1) Aligning basic axioms was less effective or even harmful to the model, which verifies Proposition 2 (Uniformity). (2) Aligning only a part of conditional axioms $\mathbb{A}_{E_h|r_i}, \mathbb{A}_{E_t|r_i}$ that describe entity feature distributions conditioned on relation representations was significantly helpful for the model. (3) Additional improvement was observed on the model with the full conditional axioms. Note that, the improvement from "Partial" to "Full" was not as significant as that from "Basic" to "Partial". This is because that the conditional triple axioms mainly describe the axioms between relations (see Appendix A). Due to the sampling strategy of the existing datasets, the number of relations is relatively small. Few correlated relation pairs exist in the datasets, resulting in limited improvement with conditional triple neural axiom alignment.

## 4.4 FURTHER ANALYSIS ON THE BOUND

We have shown that the discrepancy between each underlying aligned pair is bounded by $\epsilon$ associated with $\lambda$, in Section 2. But we still expect to obverse empirical statistics to verify this point. To this end, we manually split the entities into two group: (1) long-tail entities, which are disconnected to seeds and have at most two neighbors; (2) popular entities, the remaining. We draw the histgrams of alignment rankings w.r.t. respective groups based on the EEA model SEA. From Figure 3, we can find that the proportion of the inexact alignments (i.e., ranking $> 5$) for long-tail entities is evidently larger than that of popular entities, especially for the bins $[50, 100]$. This verified that the long-tail entities are less constrained compared to those popular entities. Furthermore, with NeoEA, the rankings of those long-tail entities were also improved more significantly compared with those of popular entities, which empirically proved that NeoEA tightened the representation discrepancy of those entities that were less restrained. We report the average ranking improvement on four datasets (V1) in Table 3, which shows consistent observations.

## 5 CONCLUSION

In this paper, we proposed a new approach to learn representations for entity alignment. We proved its expressiveness theoretically and demonstrated its efficiency by conducting experiments on the latest benchmarks. We observed that four state-of-the-art EEA methods gained evident improvements with NeoEA. Finally, we showed that the proposed conditional neural axioms are the key to improve the performance of current EEA methods.

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

## A    PROOF FOR THEOREM 1

*Aligning the conditional neural axioms minimizes the discrepancies of two KGs at ontology level.*

*Proof.* We split the proofs by different axioms referred to OWL2 (Baader et al., 2005).

**ObjectPropertyDomain | ObjectPropertyRange.** The proof for ObjectPropertyDomain has been presented in Section 3, and that for ObjectPropertyRange is similar.

**ReflexiveObjectProperty | IrreflexiveObjectProperty.** If we say that a relation $r_i$ is reflexive, it must satisfy

$$\forall (e, r_i, e') \in \mathcal{T}, \ (e, r_i, e) \in \mathcal{T}, \tag{16}$$

which means each head entities of $r_i$ must be connected by $r_i$ to itself. The above rule suggests that we can align such axioms by minimizing the discrepancy between triple distributions conditioned on relation $r_i$, i.e., aligning $\mathbb{A}^1_{E_{(h,t)}|r_i}$ with $\mathbb{A}^2_{E_{(h,t)}|r_i}$. The similar to IrreflexiveObjectProperty axiom.

**FunctionalObjectProperty | InverseFunctionalObjectProperty.** We first introduce FunctionalObjectProperty axiom. It compels each head entity $e$ connected by relation $r_i$ to have exact one tail entity, implying the following rule:

$$\forall (e, r_i, e') \in \mathcal{T}, \forall e'' \in \mathcal{E}, \ (e, r_i, e'') \notin \mathcal{T}. \tag{17}$$

Evidently, the above rule is also related to the triple distribution conditioned on $r_i$. The similar to the InverseFunctionalObjectProperty axiom.

**SymmetricObjectProperty | AsymmetricObjectProperty.** The first axiom can state a relation $r_i$ is symmetric, that is,

$$\forall (e, r_i, e') \in \mathcal{T}, \ (e', r_i, e) \in \mathcal{T}. \tag{18}$$

It is also related to the triple distributions referred to $r_i$, implying that directly aligning $\mathbb{A}^1_{E_{(h,t)}|r_i}$ with $\mathbb{A}^2_{E_{(h,t)}|r_i}$ is sufficient to minimize the difference. The proof for AsymmetricObjectProperty axiom is similar.

**SubObjectPropertyOf | EquivalentObjectProperties | DisjointObjectProperties | InverseObjectProperties.** We show that these axioms also define rules related to triple distributions condionted on relations. We start from SubObjectPropertyOf, which can state that relation $r_i$ is a subproperty of relation $r_j$ (e.g., "hasDog" is one of the subproperties of "hasPet"). We formulate it as:

$$\forall (e, r_i, e') \in \mathcal{T}, \ (e', r_j, e) \in \mathcal{T}. \tag{19}$$

To algin the potential SubObjectPropertyOf axioms between two KGs, we can respectively align $(\mathbb{A}^1_{E_{(h,t)}|r_i}, \mathbb{A}^2_{E_{(h,t)}|r_i})$ and $(\mathbb{A}^1_{E_{(h,t)}|r_j}, \mathbb{A}^2_{E_{(h,t)}|r_j})$, such that the joint one $(\mathbb{A}^1_{E_{(h,t)}|r_i,r_j}, \mathbb{A}^2_{E_{(h,t)}|r_i,r_j})$ will also be aligned.

Similarly, if $r_i$ and $r_j$ are equivalent, we can interpret the axiom as

$$\forall (e, r_i, e') \in \mathcal{T}, \ (e', r_j, e) \in \mathcal{T}; \quad \forall (e, r_j, e') \in \mathcal{T}, \ (e', r_i, e) \in \mathcal{T}. \tag{20}$$

If they are disjoint, the corresponding rule will be

$$\forall (e, r_i, e') \in \mathcal{T}, \ (e, r_j, e') \notin \mathcal{T}. \tag{21}$$

If they are inverse to each other, the rule is

$$\forall (e, r_i, e') \in \mathcal{T}, \ (e', r_j, e) \in \mathcal{T}. \tag{22}$$

**TransitiveObjectProperty.** We show that this axiom are also related to triple distributions conditioned on $r_i$. Supposed that a relation $r_i$ is transitive, then one can derive the following rule:

$$\forall (e, r_i, e') \in \mathcal{T} \& (e', r_i, e'') \in \mathcal{T}, \ (e, r_i, e'') \in \mathcal{T}, \tag{23}$$

which means we can align the potential TransitiveObjectProperty axioms via minimizing the distribution discrepancy between $\mathbb{A}^1_{E_{(h,t)}|r_i}$ and $\mathbb{A}^2_{E_{(h,t)}|r_i}$.

$\square$

## B    NEURAL ONTOLOGY ALIGNMENT

There exist several ways to implement the neural ontology alignment module. But it must satisfy two requirements: (1) Suitability. As mentioned in Section 2, different EEA benchmarks have different assumptions about the relation alignments, which means that in some cases we cannot obtain all aligned relation pairs in advance. (2) Efficiency. As there may exist millions of triples, we should consider the efficiency of performing the conditioning operations.

Therefore, in our implementation, we do not perform pair-wise sampling, either the pair-wise neural axiom alignment. We share the parameters of Wasserstein-distance critic only in each *type* of neural axiom, which saves parameters and avoid the situation that some relations only have a small number of corresponding triples. Furthermore, this also allows us to perform fast mini-batch training by aligning the axioms belonging to the same type in one operation. Given the sample KGs $\mathcal{G}'_1 = (\mathcal{E}'_1, \mathcal{R}'_1, \mathcal{T}'_1)$, $\mathcal{G}'_2 = (\mathcal{E}'_2, \mathcal{R}'_2, \mathcal{T}'_2)$, the corresponding batch loss is:

$$
\begin{aligned}
\mathcal{L}_{sep} = \mathcal{L}_{\mathbb{A}_{E'}} + \Big( \sum_{r' \in \mathcal{R}'_1} \mathbb{E}_{\mathbb{A}_{E'_h|r'}} [f_{h|r}(\mathbf{e}|\mathbf{r}')] - \sum_{r' \in \mathcal{R}'_2} \mathbb{E}_{\mathbb{A}_{E'_h|r'}} [f_{h|r}(\mathbf{e}|\mathbf{r}')] \Big) \\
+ \Big( \sum_{r' \in \mathcal{R}'_1} \mathbb{E}_{\mathbb{A}_{E'_{h,t}|r'}} [f_{h,t|r}(\mathbf{e}_h, \mathbf{e}_t|\mathbf{r}')] - \sum_{r' \in \mathcal{R}'_2} \mathbb{E}_{\mathbb{A}_{E'_{h,t}|r'}} [f_{h,t|r}(\mathbf{e}_h, \mathbf{e}_t|\mathbf{r}')] \Big),
\end{aligned}
\tag{24}
$$

where $\mathcal{L}_{\mathbb{A}_{E'}}$ is the basic axiom loss under the sampled KGs. $f_{h|r}$ and $f_{h,t|r}$ are the critic functions of the two types of neural axioms, respectively. The loss $\mathcal{L}_{batch}$ will approximate to that in a pair-wise calculation when the batch size is considerably greater than the number of relations. We take the second term of the above equation as example. For pair-wise estimation, the corresponding loss should be:

$$
\begin{aligned}
\sum_{(r_1, r_2) \in \mathcal{S}_r} (\mathbb{E}_{\mathbb{A}_{E'_h|r_1}} [f_{h|r}(\mathbf{e}|\mathbf{r}_1)] - \mathbb{E}_{\mathbb{A}_{E'_h|r_2}} [f_{h|r}(\mathbf{e}|\mathbf{r}_2)]) \\
= \sum_{(r_1, r_2) \in \mathcal{S}_r} \mathbb{E}_{\mathbb{A}_{E'_h|r_1}} [f_{h|r}(\mathbf{e}|\mathbf{r}_1)] - \sum_{(r_1, r_2) \in \mathcal{S}_r} \mathbb{E}_{\mathbb{A}_{E'_h|r_2}} [f_{h|r}(\mathbf{e}|\mathbf{r}_2)],
\end{aligned}
\tag{25}
$$

where $\mathcal{S}_r \subset \mathcal{R}_1 \times \mathcal{R}_2$ denotes the set of all aligned relation pairs. The above equation suggests that the pair-wise loss is based to the respective relation sets of two KGs, not constrained by each pair of aligned relations. Oftentimes, the number of relations is much smaller than the number of sampled triples in one batch, which means that $\mathcal{R}'_1, \mathcal{R}'_2$ in Equation 24 can cover a large proportion of elements in the full relation sets $\mathcal{R}_1, \mathcal{R}_2$. Therefore, we used $\mathcal{L}_{sep}$ to approximate the pair-wise loss in the implementation.

