# OpenReview forum: "Towards Principled Representation Learning for Entity Alignment"
_ICLR.cc/2021/Conference — Reject_

### Official Review · AnonReviewer2 · 2020-10-28
**Review for this paper**

**Rating:** 5
**Confidence:** 3

**Review:**

Summary:
The paper proposes NeoEA, an approach that further constrains KG embedding with ontology knowledge. The paper first tries to summarize the existing embedding-based entity alignment methods, stating that most of the methods choose TransE as scoring functions. But their embedding features are not aligned well compared to the neural-based or composition-based loss function. The paper, therefore, solves this problem by developing a new NeoEA architecture which shows that adding a KG-invariant ontology knowledge can minimize such difference. The experiment shows the new constraints can improve state-of-the-art baselines.

Strengths:
+ The idea of using ontology constraint as an additional loss is new. The paper shows significant improvements when combining the newly designed loss with state-of-the-art baselines.
+ The overall paper is clear to understand.

Weaknesses:
- The paper doesn't have a related work section. Figure 1 is a little bit messy. Especially for figure 1d, it would be better to make the figure a little bit larger. Moreover, it's unclear which color corresponds to the first KG, making readers confused. Section 2 should be merged into the introduction section.
- Some of the words are confusing such as neural axioms. The neural ontology alignment (which is stated in the appendix figure) is much more clear than the current Conditional Neural Axioms. Section 3 is not well-organized. The theorems and axioms should be propositions or some hypothesis. The usage of those words is a little bit wield here. The proof in the appendix is also not well defined.
- The experiment section is too short and not very informative. It would be better to include more comprehensive analysis such as providing some visualization before and after the new loss etc.

**Post-Rebuttal:
- I appreciate that the authors have conduct revisions on the current version. However, I think the current paper is probably still not strong enough for ICLR.

---

> ### Author Response · Authors · 2020-11-16
> **Response to Reviewer #2**
>
> Thanks for your detailed comments. The following are the answers to your questions.
>
> 1\. The paper doesn't have a related work section. Section 2 should be merged into the introduction section.
>
> We introduce the related work in Sect. 2.1 (for the existing EEA methods) and Sect. 3.1 (for domain adaptation), respectively. The reason why we did not organize it as an independent section is that we want to provide a seamless reading experience.
>
> 2\. Figure 1 is a little bit messy, it would be better to make the figure a little bit larger. it's unclear which color corresponds to the first KG.
>
> We have scaled it up and added a clarification about the colormap in the revision, as it allows one additional page now. Thanks for your suggestion.
>
> 3\. Section 3 is not well-organized. Some of the words are confusing such as neural axioms. The neural ontology alignment (which is stated in the appendix figure) is much more clear than current Conditional Neural Axioms. Section.
>
> Thanks for your suggestion. We have correspondingly restructured this section, by adding a subsection to illustrate neural ontology alignment, revising the usage of some words, etc. We use the word ``axiom’’ because the function of conditional distributions is similar to that of the axioms in ontologies (please see the penultimate paragraph of Introduction). We have no intention of emphasizing the mathematical significance. We have further clarified this part in the revision.
>
> 4\. The experiment section is too short and not very informative. It would be better to include more comprehensive analysis such as providing some visualization before and after the new loss etc.
>
> The experiment section looks short because we use many pages for the main content with theoretical analysis to make it self-contained. However, our experiment is still informative. We conducted evaluation on standard benchmarks. We also had a detailed ablation study on four datasets to support our main idea. Furthermore, we replaced some figures (like Figure 2) with a single table to save space, and the virtualization before and after applying NeoEA is shown in Figure 1a and 1b.

---

### Official Review · AnonReviewer3 · 2020-10-28
**Interesting but not clear paper**

**Rating:** 5
**Confidence:** 4

**Review:**

Overall Comments:
Entity alignment plays an important role in improving the quality of cross-lingual knowledge graphs. As one of the most important solutions, embedding-based methods aim at learning a semantic space where the unique entity cross knowledge graphs can have the closest distance. Most of research focus on entity-level granular, but discard the whole picture of embedding space of cross-lingual KGs. Besides the aligned entity pairs as the labelled data, this paper extended the labelled data with the conditional neural and basic axioms, which are actually sets of randomly selected entities or entities with the same relation type. Then the final objective is to align the cross-lingual knowledge graphs by both optimizing the distance of labelled entity pairs and neural axioms.

Clarity:
The presentation and organization of this paper is very difficult to follow.  Besides the grammar and type errors, there exists many concepts that not clear, which makes it difficult to understand the main idea of this work. For example, the concept of axiom and ontology are introduced before giving a formal definition. The claimed challenges, that have not been solved well by previous works ,are not convincing enough. In the 3rd paragraph, authors argued that previous research shows very good performance, but has not made on the theoretical analysis. After reading the whole draft, it's still a big question on the given theoretical analysis of this work. Taking the theorem 1 as example, it's more like a justification but not a theorem to show the connection between the proposed axiom and "ontology". Ontology provides an empirical structure to organize and classify the entities in the KGs. Its structure will be changed along with the KG in hand. From this paper, I can not find the connection between relation type alignment loss and the ontology. Please pay more attention to the writing and organization. It's an interesting work but not ready.

Questions for Rebuttal:
1. How to build the relation seed in this work? Are they labelled manually? If so, it will have flexibility issue for dealing multiple KGs.

2. Please compare to a recent proposed method [1] which also optimizes the distance between a group of entities from cross-lingual KGs. Different from this work, it's not condition on the relation type, but based on a randomly sampled group of entities.

Minor Comments:

1. In the paragraph around the Equation (6), the e_x^2 should be e_y^2.

2. Figure 3 shows the overall architecture of the proposed method. It should appear in the main content.

References:

[1] Pei, Shichao, Lu Yu, and Xiangliang Zhang. "Improving cross-lingual entity alignment via optimal transport." International Joint Conferences on Artificial Intelligence Organization, 2019.

---

> ### Author Response · Authors · 2020-11-16
> **Response to Reviewer #3**
>
> Thanks for your helpful comments. We address your concerns below:
>
> 1\. How to build the relation seed in this work? Are they labelled manually?
>
> The benchmarks used in this paper do not contain any relation seed, so we have to adapt NeoEA to this setting. In Appendix B (Eq. 25), we prove that the conditional loss can be approximated without the information of aligned relation pairs, if the batch-size is large enough.
>
> 2\. Please compare to a recent proposed method [1]
>
> Thanks for your suggestion. It is an impressive paper. We have cited it in our revision. In short, OTEA is one of the specific EEA methods that learn projection matrices to map entities in one KG to another, while our method tries to provide a general solution to facilitate EEA methods by aligning the entity representation distributions of two KGs. Hence, our method NeoEA and existing EEA methods including OTEA have different goals. We are considering adapting NeoEA to OTEA in future work, which would be interesting.
>
> 3\. Minor comments
>
> \- Typos.
> Thanks, we have corrected them.
>
>
> \- Figure of architecture.
> Thanks for your suggestion. Due to the limitation of the paper length, we put it in the appendix when submitting the paper. We have moved it into the main text.

---

### Official Review · AnonReviewer4 · 2020-10-28
**Empirically good, but the methodological contribution is limited.**

**Rating:** 5
**Confidence:** 3

**Review:**

--- Overall ---

This paper proposes an entity alignment framework that leverages the dependencies between entities and relations (reminiscent of TransR [Lin Y, et al. AAAI '15]) to further refine the results of conventional embedding-based alignment approach.

Merits: The proposed framework is shown to be effective in improving the performance of baseline models.

Weaknesses: (1) the methodological contribution is limited. (2) the theoretical explanation part is trivial and contributes little scientific knowledge.


--- major comments ----

1.	The bound in Eq.5 seems meaningless since the assumption (i.e., one relation and one neighbour) on which the bound basis is too idealistic to meet in practice.
2.	The explanations about the behaviour of embedding-based entity alignment (in both section 2.2 and section 3.1) are straight-forward and trivial, thus contribute little knowledge.
3.	In my point of view, section 2.1 and section 3.1 are too lengthy. It would be better to highlight the most important part i.e., loss function, while avoid emphasising too much on the detailed definitions and examples.


--- minor comments---
There are some typos and grammar mistakes, need to be proof-read carefully (e.g., “e_x^2” -> “e_y^2” in the paragraph just above Eq.6; “take X for example”-> “take X as an example”).

---

> ### Author Response · Authors · 2020-11-16
> **Response to Reviewer #4**
>
> Thanks for your detailed comments. We hope that the following explanations can erase your concerns.
>
> 1\. The bound in Eq.5 seems meaningless since the assumption (i.e., one relation and one neighbour) on which the bound basis is too idealistic to meet in practice.
>
> Indeed, Eq.5 is deduced from the simple assumption. In practice, entities have more than one neighbors and thus more restrictions are posed on them. The bound between two potentially-aligned entities, therefore, should be tighter than that under this assumption. That is why we used $\epsilon$ rather than $\lambda$ in Eq.5. We therefore come to the conclusion that the real bound is proportional to $\lambda$.
>
> 2\. The explanations about the behaviour of embedding-based entity alignment (EEA) is straight-forward and trivial.
>
> We agree that the explanations about embedding-based entity alignment (Sect. 2.2) as well as conventional domain adaptation (Sect. 3.1) are straight-forward. For Section 2.2, we want to provide a general and reasonable explanation for EEA, rather than the analysis towards a specific method. In Section 3.1, we want to briefly introduce what the domain adaptation is and how it works. Hence, it can be viewed as a part of related works.
>
> 3\. Section 2.1 and section 3.1 are too lengthy.
>
> As Reviewer2 mentions, our paper does not have an independent section for related work. Instead, we introduce them in Sect. 2.1 (for the existing EEA methods) and Sect. 3.1 (for domain adaptation), respectively. These two subsections are supposed to provide readers with a comprehensive analysis about recent studies, and meanwhile, to pursue a seamless reading experience. We will further improve these sections in the revision.
>
> 4\. Typos.
>
> Thanks, we have corrected them.

---

### Official Review · AnonReviewer1 · 2020-10-29
**Well justified EEA model**

**Rating:** 8
**Confidence:** 3

**Review:**

In the paper, the authors propose to minimize the discrepancy between pairs of (conditional) neural axioms to align the embedding spaces of different KGs. This method is justified by the authors' study of all kinds of OWL2 properties. The author also studied the influence of margin $\lambda$ on less constrained/long-tail entities. The authors conducted experiments by adding the proposed model on top of the best models for entity alignment. The results are mixed, but the proposed model improves the SEA and RDGCN consistently.

Reasons to accept:
1. This paper provides a theoretic point of view of the entity alignment task, which was mostly studied in empirical methods. The idea to align the axioms by minimizing Wasserstein distance is well-justified.
2. The experiment results are in favor of the intuitions.
2. The method described in this paper can be in principle adapted to any previous and future EEA scoring functions.

Reasons to reject:
The idea of using adversarial training to align spaces, especially cross-lingual spaces, is based on the assumption of the large overlap between KGs. For KGs that are on very different domains, this method may include errors, as two heterogeneous KGs do not naturally fit in one unified space. The influence of overlap on this method is not well-studied. All of KGs used in the experiments are general domain KGs.

---

> ### Author Response · Authors · 2020-11-16
> **Response to Reviewer #1**
>
> Thanks for your kind comments and suggestions. For your concern about the small overlap between heterogeneous KGs, we believe that it remains a challenge in the entity alignment area. Under the general entity alignment setting, current EEA methods usually assume that the aligned entities should share similar semantics to enable knowledge transfer. But sometimes, as you have mentioned, we may also want two KGs specific to different domains to be aligned, such that the knowledge can be really enriched rather than just linked. We are pleased to take account of it in our future work.

---

### Decision · Program_Chairs · 2021-01-07
**Final Decision**

**Decision:**

Reject

**Comment:**

The authors study the problem of augmenting embedding-based entity alignment in knowledge graphs (KG) through the use of joint alignment with deduced neural ontologies (more specifically, alignment of the KG 'neural' axioms). Motivated by the observation that the representation between two potentially aligned entities must be bound by a minimal margin, which can be problematic when there are many potential alignments, they propose aligning neural axioms by Wasserstein distance-based loss between learned entity embeddings conditioned on the relation embeddings. Experiments are conducted on OpenEA against multiple strong baselines -- showing that adding the ontology alignment to these baselines improves the results.

== Pros ==
+ The addition of aligning (conditional) ontologies is ostensibly novel.
+ For KGs with sufficient entity/relation overlap, the proposed NeoEA method is applicable.
+ NeoEA has been shown empirically to improve many SoTA methods.

== Cons ==
- While the theoretical justification is a welcome motivation, the reviewers did not find the theoretical arguments significant nor convincing.
- Overall, the narrative needs work to make the paper more self-contained and approachable for a broader range of readers. The reviewers (and myself) found many concepts and statements somewhat confusing and needing clearly context and contrast with existing works.

Evaluating along the requested dimensions:
- Quality: Conceptually, the core idea is interesting, well-motivated, original, and ostensibly effective. Empirically, NeoEA is shown able to improve upon several strong baseline (underlying) methods.  I believe that all of the reviewers find the work is interesting and promising. However, there were continuing concerns the strength/value of the described theory; it isn't clear if stronger theory isn't possible or if this just hasn't been fleshed out.
- Clarity: Most of the reviewers (and myself) found the paper difficult to follow as a self-contained work in terms of concepts, clear definitions (e.g., \mathcal T isn't defined early on) and the actual applicability of the theory. The figures help, but even these need some work. A related work section (or more structured presentation of related work) might be clarifying along with running examples and a more unifying math presentation that captures existing and proposed work. After thinking about this more, it is actually a relative simple (in a good way) and clever idea. However, it took several readings and readings of related work to get there. Additionally, the fact that all of the reviewers were concerned about different limitations is concerning wrt clarity. Appendix B helps a bit and I believe can also be put into the main paper.
- Originality: As best as the reviewers and I can tell, we haven't seen this method applied to entity alignment despite this being a relatively mature subfield.
- Significance: The consensus seems to be that the approach could be a notable contribution to an important area. However, it also appears that most of the reviewers don't feel the paper is ready for publication at a top-tier venue yet.

As stated throughout this meta-review, there are several aspects to like about this work including the originality of the idea, strong motivation, and good empirical results. However, we all agreed that the paper isn't quite ready in its current form -- thus, I presently recommend reject for this submission.